# Effect of Different Phosphates on Pyrolysis Temperature-Dependent Carbon Sequestration and Phosphorus Release Performance in Biochar

**DOI:** 10.3390/molecules28093950

**Published:** 2023-05-08

**Authors:** Tianxia Bai, Wenge Ma, Wenhui Li, Jinling Jiang, Jiamin Chen, Rui Cao, Wenjie Yang, Dan Dong, Tingwu Liu, Yonggang Xu

**Affiliations:** 1Jiangsu Key Laboratory for Eco-Agricultural Biotechnology around Hongze Lake, Collaborative Innovation Center of Regional Modern Agriculture & Environmental Protection, Huaiyin Normal University, Huai’an 223300, China; baitianxia2008@163.com (T.B.);; 2School of Chemistry and Chemical Engineering, Huaiyin Normal University, Huai’an 223300, China

**Keywords:** biochar, carbon retention, corn stalk, phosphorus, stability

## Abstract

Carbon sequestration is the primary function of biochar. Hence, it is necessary to design biochar with high carbon (C) retention and low C loss. In this study, three P compounds, including KH_2_PO_4_, Ca(H_2_PO_4_)_2_, and NH_4_H_2_PO_4_, were premixed with corn stalk (1:4, *w*/*w*), aiming to produce biochars (CSB+K, CSB+Ca, and CSB+N) with high C sequestration and slow release of P at three temperatures (300, 500, and 700 °C). The addition of all P sources obviously increased C retention, with the order of NH_4_H_2_PO_4_ (65.6–83.5%) > Ca(H_2_PO_4_)_2_ (60.4–78.2%) > KH_2_PO_4_ (50.1–76.1%), compared with the pristine biochar (47.8–73.6%). The addition of Ca(H_2_PO_4_)_2_ and KH_2_PO_4_ led to an increase in aromaticity and graphitization, as evidenced by H/C, FTIR, Raman and XPS analysis, whereas an opposite result occurred on CSB+N. Furthermore, all three phosphates reduced C loss of biochars with H_2_O_2_ oxidation, and CSB+Ca showed the best effect. Ca(H_2_PO_4_)_2_ and KH_2_PO_4_ pretreated biochars had higher resistance to K_2_Cr_2_O_7_ oxidation and thermal treatment. In contrast, the C loss of NH_4_H_2_PO_4_-added biochar at 500 and 700 °C with K_2_Cr_2_O_7_ oxidation was increased by 54% and 36%, respectively. During the pyrolysis process, Ca(H_2_PO_4_)_2_ was transformed into insoluble Ca_2_P_2_O_7_, leading to the lowest P release rate of CSB+Ca. This study indicates that co-pyrolysis of corn stalk and Ca(H_2_PO_4_)_2_ is optimal for increasing C retention, enhancing C stability and improving slow-release performance of P regardless of pyrolysis temperature.

## 1. Introduction

Human development accelerates agricultural production, and a large amount of biomass waste is generated, which was estimated as about 700 Mt of crop residues and 900 Mt forest residues annually in China [1]. The carbon (C) that exists in biomass waste is generally liberated to the atmosphere via biodegradation and combustion, and thus became a main source of greenhouse gases, contributing to climate change and global warming [2]. A viable and effective method for sequestrating C in biomass waste need to be developed. It has been proven that pyrolytic conversion of biomass waste into biochar is a prospective method for C sequestration [3]. In addition, when being applied into soil, biochar has multiple additional benefits, such as improving soil quality [4] and stabilizing contaminants [5]. However, there are still several barriers that limit the wide practical application of biochars.

On the one hand, C retention and stability of biochar are important for the purpose of C sequestration, which mainly depend on the feedstock types and pyrolysis conditions [4]. Generally, C in biochar becomes more stable with increasing pyrolysis temperature, whereas more C is also lost [6,7]. Thus, it is urgent to create biochar with high C retention and stability. Many studies have found that exogenous additives could reach this purpose [8,9,10]. For instance, phosphate has been shown to better enhance C retention than other minerals because of the formation of stable P complexes (i.e., C−O−P or C-P) [11,12,13]. Moreover, Xu et al. [14] found that Ca(H_2_PO_4_)_2_ could boost the chemical stability of C in sludge-derived biochar, but it was only effective for biochar pyrolyzed at 600 °C, and it was concluded that C-O-PO_3_ and C-PO_3_ groups were formed at high temperature, which cannot be decomposed by H_2_O_2_ and K_2_Cr_2_O_7_. In contrast, Carneiro et al. [15] reported that the addition of H_3_PO_4_ or H_3_PO_4_−MgO (1:1 molar ratio, equivalently MgHPO_4_) at 500 °C greatly improved C retention and thermal stability of biochar derived from coffee husk and poultry litter, but a lower chemical stability (H_2_O_2_ and K_2_Cr_2_O_7_ oxidation) of C was observed in these biochars, most likely due to their higher specific surface area and smaller crystallite size. Despite that research has been performed on C retention and stability of the resultant biochars after phosphate addition, the contrast effect of different levels of phosphate addition as well as pyrolysis conditions on C sequestration of biochar is rarely reported.

On the other hand, biomass derived-biochar cannot be directly used as a fertilizer due to low nutrient contents. Co-pyrolysis of biomass and P-containing materials has been proved to be a novel strategy that has been used in the production of high-performance biochar-based slow-release fertilizers [16,17,18]. Zhao et al. [12] firstly synthesized biochar-based fertilizer via co-pyrolysis of sawdust or switchgrass and two P sources (triple superphosphate and bone meal), and revealed that these biochars show a weaker release performance of P compared to two P sources over 120 h due to the formation of stable C-P complexes and insoluble Ca_2_P_2_O_7_. However, Lustosa Filho et al. [19] found that H_3_PO_4_, NH_4_H_2_PO_4_ and Ca(H_2_PO_4_)_2_ composite biochar released 99.6%, 86.0%, and 37.0% of total P, whereas released P was drastically reduced to 31.6%, 28.9%, and 9.1% after MgO addition, which resulted from the formation of Mg_2_P_2_O_7_ or Ca_10_(OH)_2_(PO_4_)_6_. The above results indicate that the formation of insoluble mineral mainly contributes to the slow release performance of P in biochar. It also means that enhanced C retention and slow release of P are contradictory. Therefore, it is essential to find a balance.

In this study, three kinds of phosphates (KH_2_PO_4_, Ca(H_2_PO_4_)_2_, and NH_4_H_2_PO_4_) were selected as additives due to their high solubility, and thus, they had a better effect on C retention than less soluble P-bearing materials. The main objectives of this study were to (1) evaluate the influence of these phosphates on C sequestration capacity by assessing C retention during pyrolysis and C stability in biochar derived from corn stalk (CS); (2) investigate the kinetic release of P from the composite biochars; (3) explore the inherent mechanisms of the above effects basing on the analysis of the C microstructure and P form.

## 2. Results and Discussion

### 2.1. The Physical and Chemical Properties of Biochars

The yield of biochar drastically decreased from 54.41% to 35.13% with increasing pyrolysis temperature from 300 to 500 °C, predominately attributing to the thermal decomposition of lignocellulose in CS [20]. Only a slight reduction of biochar yield from 35.13% to 32.18% occurred when the pyrolysis temperature was raised from 500 to 700 °C. The addition of three phosphates obviously increased the yield rates of biochar, following the order of CSB+N > CSB+Ca > CSB+K, as the result of a lower decomposition rate of phosphate than CS, whereas the ash contents showed a reverse trend, indicating that CSB+N had lower oxidative stability than CSB+K and CSB+Ca [8]. Compared to CSB, lower VS contents were found in CSB+K and CSB+Ca, while an opposite trend was observed in CSB+N, except for CSB300+N, likely resulting from the decomposition of either organic fractions of the C skeleton or some combination of C and additives [8]. Unexpectedly, the FC contents, representing the stable C in biochar, were lower in all P-modified biochars than CSB, because of the high ash contents or VS fractions in them.

In comparison to CSB (pH = 7.51, 9.23, and 10.17 for CSB300, CSB500, and CSB700, respectively), the pH value of all P-modified biochars decreased due to the acidity of phosphate and its thermal production (see Appendix A), following the order of: CSB+K (6.64–7.82) > CSB+Ca (3.55–4.04) > CSB+N (2.38–2.47). Compared to CSB, less C, N, H and more O were found in all P-modified biochars except for N in CSB+N. Compared to C content, H content was more reduced by the introduction of phosphate, probably because of the hydration reaction. Thus, the H/C ratios were significantly lower in all P-doped biochars than CSB, except for CSB300+K and CSB300+N, and the sequence of reduction is CSB+Ca > CSB+K > CSB+N, indicating enhanced aromaticity.

### 2.2. Carbon Retention of Biochar

#### 2.2.1. Effect of P Addition and Pyrolysis Temperature on C Retention of Biochar

As shown in Figure 1, C retention in CSB was sharply reduced from 73.64% of CSB300 to 50.89% of CSB500, whereas it gradually decreased to 47.77% when the temperature reached 700 °C. Although the C contents of biochars were reduced after the addition of phosphates (Table 1), more C was significantly retained in CSB+N and CSB+Ca at all pyrolysis temperatures. It was interesting that the enhancement of C retention in CSB+N and CSB+Ca quickly increased at first and then gradually with the increasing pyrolysis temperature. For example, adding Ca(H_2_PO_4_)_2_ in CS promoted C retention by 6.21%, 22.19%, and 26.48% for 300, 500 and 700 °C, respectively. The underlying mechanisms for the enhancement and variation of C retention will be discussed in the next sections.

#### 2.2.2. Thermogravimetric Analysis

The comparison of experimental and calculated data from TG plots were made to investigate whether there was any effect of phosphate on the decomposition of CS (Figure 2). Evidently, at low temperature (<320 °C), the measured value of mass loss was lower than the theoretical value, whereas an opposite trend was observed at high temperature (>320 °C), with the order of CS+NH_4_H_2_PO_4_ > CS+Ca(H_2_PO_4_)_2_ > CS+KH_2_PO_4_, indicating that the effect of phosphate on the decomposition of CS depended on pyrolysis temperature and phosphate types. Alkali and alkaline Earth metal salts were reported to be responsible for weakening hydrogen bonds and decreasing the stabilities of hydroxyl group and glycosidic bond, resulting in the thermal decomposition occurring in advance [21], and Ca had stronger ability than K [22], as shown in Appendix A. Li et al. [23] also confirmed that the breakage of glycosidic bond and dehydration reaction were promoted, and the energy required for decomposition was decreased by NH_4_H_2_PO_4_ during the pyrolysis of rice husk, mainly due to the role of produced H_3_PO_4_ or H_4_P_2_O_7_ (Equations (1) and (2)). When the temperature exceeded 320 °C, the retained solid increased by impregnating phosphate, mainly resulting from the polycondensation reaction facilitated by K or Ca for char formation [24,25] and the phosphate esterification between phosphate and hydroxyl groups for stable C-O-P or C-P groups [11].
NH_4_H_2_PO_4_ → NH_3_ + H_3_PO_4_ (209 °C)(1)
2H_3_PO_4_ → H_4_P_2_O_7_ + H_2_O (180–280 °C)(2)

#### 2.2.3. SEM

The morphological characteristics of biochar were analyzed by SEM (Appendix A). With increased temperature, the structure of CSB became crumbly and rough. In contrast, the structure of CSB+K became more collapsed, and many tiny particles were found in CSB700+K. In contrast, some bigger irregular particles were presented on the surface of CSB+Ca and CSB+N (Appendix A). Moreover, many cracked bubbles appeared in CSB+N (Appendix A). 

#### 2.2.4. XRD

XRD analysis showed that (KPO_3_)_n_ was present in CSB+K (Appendix A), resulting from the dehydration and condensation of KH_2_PO_4_ (Equation (3)). In the case of CSB+Ca, most of the Ca(H_2_PO_4_)_2_ was decomposed into insoluble Ca_2_P_2_O_7_ and H_3_PO_4_ (Equation (4)), and the latter further reacted with KCl to form KH_2_PO_4_ (Equation (5)). Some diffraction peaks related to KH_2_PO_4_ were also observed in CSB+N. It was interesting that the SiO_2_ peak appeared in CSB700 and CSB700+K, whereas no crystalline SiO_2_ was detected in CSB700+Ca and CSB700+N, indicating that it might react with Ca(H_2_PO_4_)_2_ and NH_4_H_2_PO_4_ at high temperature. Vaimakis and Sdoukos [26] reported that the reaction product between Ca(H_2_PO_4_)_2_ and SiO_2_ was Si(HPO_4_)_2_ at lower temperatures, and it decomposed to SiP_2_O_7_ at higher temperatures, in which both are amorphous. Especially, a sharp peak at 2θ = 26.6 was observed in CSB700+N and CSB700+Ca, most likely corresponding to C_3_-P=O linkages in the lattice and on the edges of graphite-like crystallites [8,27].
nKH_2_PO_4_ → (KPO_3_)n + nH_2_O(3)
2Ca(H_2_PO_4_)_2_ → Ca_2_P_2_O_7_ + 2H_3_PO_4_ + H_2_O(4)
H_3_PO_4_ + KCl **↔** KH_2_PO_4_ + HCl(5)

#### 2.2.5. FTIR

TheFTIR spectra of biochars are presented in Figure 3. After phosphate addition, a new peak at approximately 1080 cm^−1^, attributing to the stretching vibration of C-O-P [12] or to the symmetrical vibration in polyphosphate chain P−O−P [18], was observed in all P-doped biochars. Other researchers also reported the existence of C-O-P groups in P-doped biochars, which could work as physical barriers against carbon decomposition [8,28]. Additional peaks at 1262, 880, 759 and 677 cm^−1^, attributing to the asymmetrical stretching vibration of O–P–O as well as the asymmetrical and symmetric stretching vibration of P-O-P in (PO_3_^−1^)_n_, respectively, were observed in CSB+K [29]. For CSB+Ca, new peaks at 1140, 1000, and 720 cm^−1^ appeared, which corresponded to the asymmetrical and symmetric stretching vibrations of O-P-O and the asymmetrical and symmetric stretching vibrations of P-O-P in γ-CaP_2_O_7_, respectively [30]. Moreover, the bands at 557, 533, 495, and 452 cm^−1^ in CSB500+Ca and CSB700+Ca were also related to the blending of O-P-O in γ-CaP_2_O_7_ [30]. The above results supported the existence of (KPO_3_)_n_ and CaP_2_O_7_ in CSB+K and CSB+Ca, respectively. Additionally, a new peak at around 1030 cm^−1^ appeared in CSB+Ca and CSB+K, suggesting the enhancement of aromatic C=O [31]. Especially, two new peaks at 954 and 1259 cm^−1^ that are attributed to Si-O-P [32] appeared in all CSB+N, CSB500+Ca and CSB700+Ca, respectively, supporting the presence of Si(HPO_4_)_2_ or SiP_2_O_7_.

#### 2.2.6. XPS

Appendix A and Table 2 show the XPS spectra of C_1s_ and its deconvolution results of all biochars. The C–C/C=C group was the most dominant C speciation in all biochar, and it increased from 76.92% of CSB300 to 86.21% of CSB700. Simultaneously, the relative percentages of O-containing groups (i.e., C-O, C=O, and O–C=O) declined. The relative percentage of the C–C/C=C group was less in P-doping biochars than the corresponding pristine biochar, whereas the relative percentages of C-O (or C-O-P) increased, following the order of CSB+N > CSB+Ca > CSB+K. Previous studies also reported that phosphoric acid was more favorable for the formation of phosphoester (C-O-P) than phosphates (orthophosphate, pyrophosphate, and polyphosphates) [33,34]. This result agrees very well with the performance of C retention, confirming that the formation of C-O-P contributed to improving C retention.

The inorganic (PO_3_^−^, P_2_O_7_^4−^, HPO_4_^−^, and H_2_PO^4−^, etc.) and organic phosphate groups (C_3_-P, C_2_−P−O_2_, C-P-O_3_, and C−O−P, etc.) shared similar binding energies values (132.2–134.0 and 131.3–134.4 eV, respectively) [35,36,37], and thus, the deconvolution of P_2p_ spectra could not be analyzed. Nevertheless, compared to the corresponding pristine biochar, higher binding energies of P_2p_ in the P-modified biochars also indicated the presence of C-O-P groups (Appendix A) [38]. In addition, the P2p peak of CSB+Ca and CSB+N showed an obvious displacement to higher binding energies with increasing temperature, likely due to the formation of the Si-O-P bond (135.5 eV) [39]. On the contrary, the peak of P2p in CSB+K decreased from 134.2 to 133.8 eV with the increasing temperature, due to the transformation of H_2_PO_4_^−^ (134.3 eV) into PO_3_^−^ (132.9 eV) [36], as confirmed by XRD analysis (Appendix A). 

### 2.3. Chemical Stability of C in Biochar

As shown in Figure 4a, the C loss of all P-modified biochars after H_2_O_2_ oxidation were in descending order of CSB+Ca > CSB+K > CSB+N, which were all less than those of the corresponding pristine biochars, except for CSB300+N. Although the introduction of NH_4_H_2_PO_4_ led to the best enhancement of C retention during CS pyrolysis (Figure 1), the percentage of unstable C of CSB300+N, CSB500+N, and CSB700+N increased to 95.5%, 53.0%, and 42.2%, respectively, expressed as K_2_Cr_2_O_7_ oxidation, compared to CSB300 (93.6%), CSB500 (34.5%), and CSB700 (30.1%). In contrast, CSB+Ca and CSB+K also showed stronger resistance to K_2_Cr_2_O_7_ oxidation except for CSB300+K. Raman spectra showed that the I_D_/I_G_ ratios of CSB500+K (0.85) and CSB500+Ca(H_2_PO_4_)_2_ (0.78) were lower than that of CSB500 (1.30), indicating higher graphitization and ordering of C structure [40]. This was likely because of the catalytic effects of alkali and alkaline Earth metals on biomass pyrolysis, such as the cleavage of anhydrosugar (e.g., levoglucosan) to produce more light oxygenated compounds and furan species [24,41], which serve as precursors and are prone to the formation of aromatic C through polymerization reactions [42,43]. However, the addition of NH_4_H_2_PO_4_ caused an elevation in the I_D_/I_G_ ratio (1.71) (Appendix A). XRD analysis also showed a wide peak at around 2θ = 24°, representing amorphous C diffraction in CSB+N (Appendix A) [44]. A previous study also showed that the introduction of H_3_PO_4_ was negative to the formation of an ordered and graphitic C structure [45]. Moreover, a higher surface area was found in CSB500+K and CSB500+N (Appendix A), meaning that the access of oxidants to two biochars could be enhanced, and a considerable quantity of volatiles could be trapped inside these biochars, which was readily decomposed by the oxidant [11], leading to an increase in C loss. However, no significant difference was observed between CSB500 and CSB500+Ca. The changes of combined properties led to the lowest chemical stability of CSB+N, followed by CSB+K and CSB+Ca.

As mentioned above, C retention and stability of biochars were influenced greatly by the introduction of phosphates, whereas biochar with the highest C retention ratio was not the most stable. Consequently, it is necessary to accurately evaluate C sequestration using a reasonable method. According to the study of Nan et al. [9], the C sequestration capacity of biochar can be quantified by the C content in residue after K_2_Cr_2_O_7_ oxidation. In this study, NH_4_H_2_PO_4_ had little influence on C sequestration of biochar pyrolyzed at 300 °C (4%) and 500 °C (32%), whereas the final sequestrated C in biochar produced at 700 °C was increased (38%). This phenomenon may be due to the formation of stable C_3_-P-O, as confirmed in XRD analysis (Appendix A). Among all biochars, CSB+Ca showed the highest C sequestration regardless of pyrolysis temperature, and the enhanced effect increased with rising pyrolysis temperature. These results suggested that the effect of doping phosphate on C sequestration of biochar depended on the phosphate type as well as pyrolysis temperature.

### 2.4. Thermal and Oxidative Stability of Biochar

Generally, the higher the inorganic fraction, the lower the mass loss in the TGA analysis, regardless of atmosphere, because only a small portion of inorganic material was vaporized during the process [46]. As shown in Table 1, higher ash contents were detected in all P-doped biochars. Therefore, the TGA patterns of biochars were corrected by ash content (Figure 5). Our results showed that the higher pyrolysis temperature, the greater the thermal stability of biochars, which was consist with previous studies [47,48]. The mass losses of CSB+K and CSB+Ca pyrolyzed at three different temperatures were lower than pristine biochar during the whole pyrolysis process (Figure 5a–c), and the CSB+Ca showed higher thermal stability than CSB+K. This was likely because K could catalyze the degradation of biochar C during TGA analysis [49]. In comparison, the CSB300+N had lower mass loss than CSB300 at >330 °C (Figure 5a), while CSB500+N and CSB700+N showed higher mass loss at the total range of temperature (Figure 5b,c), likely because of the decomposition of either organic fractions of biochar or further combination of biochar and additives [8]. Overall, the above results indicated that pyrolysis temperature obviously affected the thermal stability of biochar. 

It has been proven that phosphorus complexes, i.e., C–O–P or C-P groups act by stabilizing the carbon-active sites against carbon loss during heat treatment under an air atmosphere [50,51]. However, CSB+N with the highest amount of C-O-P groups showed the lowest oxidative stability at low temperature (<390 °C), which could be attributed to the loss of OH group of polyphosphoric acid via esterification [15]. Hence, the newly formed C-O-P group led to an opposite trend at high temperature (>540 °C, Figure 5d). 

### 2.5. Phosphorus Release

Total P contents in P-modified biochars (73.3 to 119.0 mg g^−1^) were much higher than those of the corresponding pristine biochars (4.7–7.9 mg g^−1^; see Table 1), making it useful as a potential P fertilizer. The release patterns of P from biochars are shown in Figure 6. The release amounts of P from CSB+K, CSB+Ca, and CSB+N were 74.9–94.46, 47.2–48.0, and 63.9–85.1 mg g^−1^ over the entire period of 240 h, accounting for 77.2–96.1%, 37.7–53.8%, and 50.4–80.3% of TP, respectively, which were much more than CSB (<4.0 mg g^−1^) (Figure 6). Overall, the P that leached from CSB+Ca was lowest among the three P-modified biochars, and this was mainly because of insoluble Ca_2_P_2_O_7_. This was confirmed by XRD of the biochar after P release, showing that abundant amounts of Ca_2_P_2_O_7_ remain in the residue (Appendix A). Additionally, the peak at around 1086 cm^−1^ (C-O-P) was still present in the P-doped biochars after 240 h (Appendix A), indicating the stability of C-O-P in aqueous solution. It was interesting that pyrolysis temperatures had little influence on the P release of CSB and CSB+Ca, whereas the ratio of it to TP decreased with increasing pyrolysis temperature (Figure 6). Despite that CSB500+N showed a higher release rate than CSB300+N, followed by CSB700+N, the ratio of it to TP at 240 h was in the order of CSB300+N (80.3%) > CSB500+N (73.3%) > CSB700+N (50.4%). The XPS pattern indicated that the fraction of C-O-P or Si-O-P groups in CSB+Ca and CSB+N increased with rising pyrolysis temperature (Appendix A). The P release via the breaking C−O−P bond needed extra energy [12], and SiP_2_O_7_ was insoluble, thus resulting in a slower release of P from biochar pyrolyzed at higher temperatures. As indicated by XPS (Appendix A), as pyrolysis temperature increased, more soluble KH_2_PO_4_ was converted into poorly soluble (KPO_3_)_n_, but fewer C-O-P group was formed in CSB+K. Thus, the trade-off between the (KPO_3_)_n_ and C-O-P group possibly led to the lowest P release ratio in CSB500+K (77.2%) compared to CSB700+K (82.4%) and CSB300+K (96.1%).

Although different amounts of P were released, all biochar exhibited similar P-release patterns, except for CSB+K. The P release from all biochars was rapid in the first 24 h, following by a continued slow release. For instance, the amount of released P from CSB700+Ca reached 34.28% and 37.71% of total P cumulatively after 24 and 240 h, respectively. As indicated in Table 3, the P release kinetics from biochars were fitted best with the Elovich and Power models. Previous studies reported that Elovich and Power equations could best describe the release pattern of P from soil, manure, or biochar [52,53]. Compared with CSB+K and CSB+N, lower values, as an initial desorption rate [54], were observed in CSB+Ca, mainly originating from the low solubility of Ca_2_P_2_O_7_. Furthermore, the medium proportion of the C-O-P group in CSB+Ca also contributed to a low initial desorption rate of P.

## 3. Materials and Methods

### 3.1. Material

Corn stalk (CS) was obtained from the agricultural field in Huaian City (Jiangsu, China). The CS was dried at 105 °C in an oven (DHG-9140, Yiheng, Shanghai, China) with moisture lower than 5%, and then, the dried samples were crushed to less than 0.2 mm [13]. KH_2_PO_4_, Ca(H_2_PO_4_)_2_, and NH_4_H_2_PO_4_ (analytical grade) were purchased from Sinoparm Chemical Reagent Company (Beijing, China).

### 3.2. Biochar Production

According to the method of our previous study [13], 50.0 g CS was completely impregnated in 200 mL solution containing 12.5 g phosphate, and these mixtures were stirred mechanically at a speed of 500 rpm for 1 h (90-1, Jingke, Shanghai, China) and then held for 24 h at room temperature. These prepared samples were dried at 105 °C for about 8 h in an oven (DHG-9140, Yiheng, Shanghai, China). All feedstocks were pyrolyzed in a tube furnace (SKGL-1200C, Jujing, Shanghai, China) by raising the temperature to 300, 500, or 700 °C at a rate of 10 °C·min^−1^, holding for 60 min under a N_2_ atmosphere (99.999%, 300 mL min^−1^). The resulting biochar without P addition (defined as pristine biochar) and those with KH_2_PO_4_, Ca(H_2_PO_4_)_2_, and NH_4_H_2_PO_4_ were named as CSBX, CSBX-K, CSBX-Ca, and CSBX-N, respectively, in which X indicated the pyrolysis temperature.

### 3.3. Thermogravimetric Analysis (TGA)

The TGA of CS with or without phosphate was investigated via a NETZSCH STA 449F3 thermogravimetric analyzer [13]. To ensure consistency of all experimental parameters, about 10 mg of samples was heated from room temperature to 800 °C at a rate of 10 °C min^−1^ under N_2_ atmosphere (99.999%, 50 mL min^−1^).

The theoretical value of mixture weight (W_cal_) is calculated as the weight sum of raw materials at a certain temperature:W_cal_ = 0.8 × W_1_ + 0.2W_2_
where W_1_ and W_2_ (wt.%) are the weights of CS and phosphate during the solo-pyrolysis process at temperature, respectively.

### 3.4. Measurement of Biochars Properties

The yield of biochar is calculated using the following equation:Biochar yield (%) = (W_T_ − W_B_)/Wo ∗ 100%
where W_T_ is the total mass (g) of the quartz boat and biochar after pyrolysis, and W_B_ and W_0_ are the mass (g) of the quartz boat and samples prior to pyrolysis, respectively.

The pH and EC values were measured with a multifunction meter (DZS-706, Leici, Shanghai, China) after shaken for 1 h at a solid–liquid ratio of 1:20 (*w*/*v*). The bulk C, H, and N contents were determined with an elemental analyzer (Vario ELII, Elementar, Hanau, Germany). The O content was calculated as follows:O content (%) = 100% − C(%) − N(%) − H(%) − ash content (%)

C retention in biochar was calculated as follow:Carbon retention (%) = TC_biochar_ × W_biochar_/(TC_CS_ × W_CS_)
where TC_biochar_ and TC_CS_ are the total C contents (%) of biochar and corn stalk, respectively, and W_biochar_ and W_CS_ are the weights (g) of biochar and corn stalk, respectively.

Ash contents and volatile solids (VS) were determined by weight loss after heating the samples at 650 °C for 4 h under air atmosphere and at 900 °C for 7 min under N_2_ atmosphere, respectively. Fixed carbon (FC) is calculated as follow:FC (%) =100 − VS (%) − Ash (%)

The specific surface area and pore size distribution of samples were measured by N_2_ adsorption isotherm at 120 °C based on the Brunauer–Emmett–Teller (BET) method (ASAP 2020, Micromeritics, Norcross, GA, USA). The crystallinities of biochar were detected by X-ray diffractometer (X’Pert3 Powder Diffractometer, PANalytical, Almelo, The Netherlands), which was operated at 40 kV and 30 mA; data were collected over the 2θ range from 5° to 80° using Cu Kα radiation with a scan speed of 10° min^−1^. The surface functional groups were obtained with Fourier transform infrared (FTIR) spectrometer (Nicolet iS50, Thermo Fisher, Waltham, MA, USA) within 4000–400 cm^−1^ under a 4 cm^−1^ resolution. The surface morphologies were acquired by scanning electron microscopy (SEM, Inspect S50, FEI, Hillsboro, OR, USA) at an operational voltage of 15 kV. The peak spectra of C_1s_ and P_2p_ of biochar were examined with an X-ray photoelectron spectroscopy (XPS) (Thermo SCIENTIFIC Nexsa, ThermoFisher, Waltham, MA, USA) using Al Ka radiation. The C microstructure of biochar was analyzed by Raman spectroscopy (LabRam Evolution HR, HORIBA, Palaiseau, France) within the range of 800–2000 cm^−1^ at 532 nm.

### 3.5. Measurement of C Stability in Biochar

Two methods (H_2_O_2_ and K_2_Cr_2_O_7_ oxidation treatments) were used to assess the chemical stability of C in biochar according to the method described by a previous study [11]. Briefly, a certain amount of biochar containing 0.10 g C was put into a glass test tube, and then, 7 mL of 5% H_2_O_2_ or 40 mL of 0.1 M K_2_Cr_2_O_7_/2 M H_2_SO_4_ solution was added. These glass test tubes were heated in an air oven and held at 80 °C for 48 h and at 55 °C for 60 h. After oxidation, the samples were washed with deionized water. The weight of residue was recorded after oven-drying at 105 °C for 6 h, and its C content was also determined using an elemental analyzer (Vario ELII, Elementar, Hanau, Germany). The chemical stability of C was expressed as the remaining ratio of C after chemical oxidation.
Chemical stability of C (%) =(W_f_ × C_f_)/(W_i_ × C_i_) × 100
where W_i_ and W_f_ are the weights (g) of sample before and after oxidation treatment, respectively; C_i_ and C_f_ are the initial and final C contents (%) of the sample, respectively.

The TGA of biochar in N_2_ and air atmosphere was used to assess the thermal and oxidation stability, respectively, in which the biochar was heated from room temperature to 800 °C at 10 °C per min.

### 3.6. Phosphorus Release from Biochar

The P release pattern was conducted according to the method of Lustosa Filho et al. [19]. Briefly, 1.0 g biochar was put into a sebc bottle (250 mL), and 200 mL of deionized water was added. These sebc bottles were shaken in a reciprocating shaker at 150 rpm at 25 °C. About 5 mL suspension was collected at 0.5, 1, 2, 4, 8, 24, 48, 120, and 240 h and filtered. The concentration of P in filtrate was measured by ICP−OES (DV2100, Perkin Elmer, Waltham, MA, USA). The kinetic analysis of P release was conducted using the following equations:First-order ln(q_0_ − q_t_)= lnq_o_ − k_1_ × t(6)
Pseudo-second t/q_t_ = 1/(k_2_ × q_0_^2^) + t/q_0_(7)
Power function lnq_t_ = lna + b × lnt(8)
Elovich q_t_ = 1/a × ln(a × b) − ln(1/b) × ln(t)(9)
Parabolic diffusion q_t_ = a + b × t^1/2^(10)
where q_t_ is the amount of released P at time t; a, b, k_1_ and k_2_ are P release constants; q_0_ is the maximum quality of released P.

### 3.7. Statistical Analysis

All experiments on the C stability and release of P were conducted in triplicate. Statistical analyses were performed using SPSS 20.0, and one-way analysis of variance (ANOVA) was used to determine differences among treatments at the 95% confidence interval with a significant level of 0.05.

## 4. Conclusions

This study demonstrated that exogenous phosphates (KH_2_PO_4_, Ca(H_2_PO_4_)_2_, and NH_4_H_2_PO_4_) influenced the multiple beneficial properties of biochar derived from corn straw in terms of C retention, stability and potential slow-release P fertilizer. Three phosphates could increase C retention in varying degrees, mainly attributing to the amount of formed C-O-P groups. Although the addition of NH_4_H_2_PO_4_ could retain more C than KH_2_PO_4_ and Ca(H_2_PO_4_)_2_, the extra retained C was stable enough to resist K_2_Cr_2_O_7_ oxidation and thermal treatments under N_2_ or air atmosphere, mainly because of increases in surface area and amorphous C. In contrast, KH_2_PO_4_ and Ca(H_2_PO_4_)_2_ could greatly enhance the chemical and thermal stability of C by promoting the formation of the aromatic structure, in which CSB700+Ca presented the best C sequestration ability. Furthermore, insoluble Ca_2_P_2_O_7_ formed on the surface of CSB+Ca, which acted as a physical barrier against the release of C-containing molecules and which led to the best potential of acting as a slow P-release fertilizer.

Only one biomass of corn straw was used in this study; more different biomass feedstocks should be tested to demonstrate the exogenous phosphates effect. In addition, C stability and the slow release of P in biochar should be assessed by soil incubation experiments because the results relate directly to the persistence of biochar in soil. Low pH (3.55–4.04) of CSB+Ca is also unsuitable for land application. More alkaline P sources (e.g., Ca(HPO_4_)_2_) should be tested for correcting the acidity of this composite biochar. Overall, from a commercial point of view, phosphate pretreatment offers a way to increase biomass C retention, stability and slower P release in biochar during pyrolysis without increasing energy consumption and production cost.

## Figures and Tables

**Figure 1 molecules-28-03950-f001:**
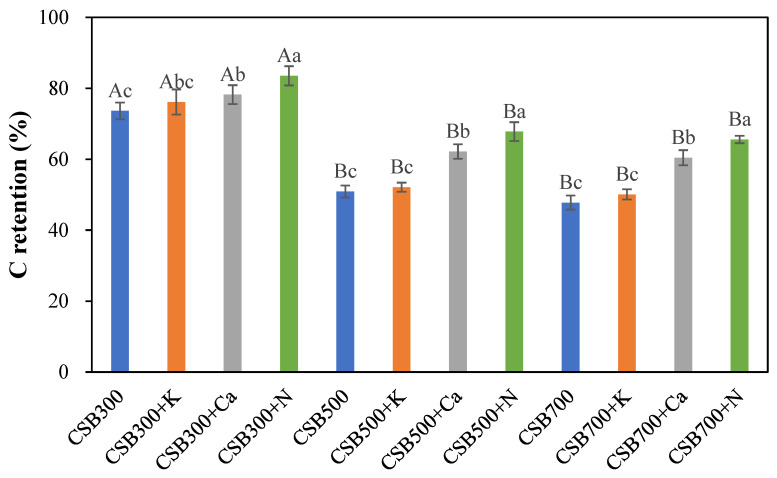
C retention of biochar with or without phosphate. The differences among different temperature treatments are represented by a big letter, and the differences among different phosphates additives are represented with a small letter.

**Figure 2 molecules-28-03950-f002:**
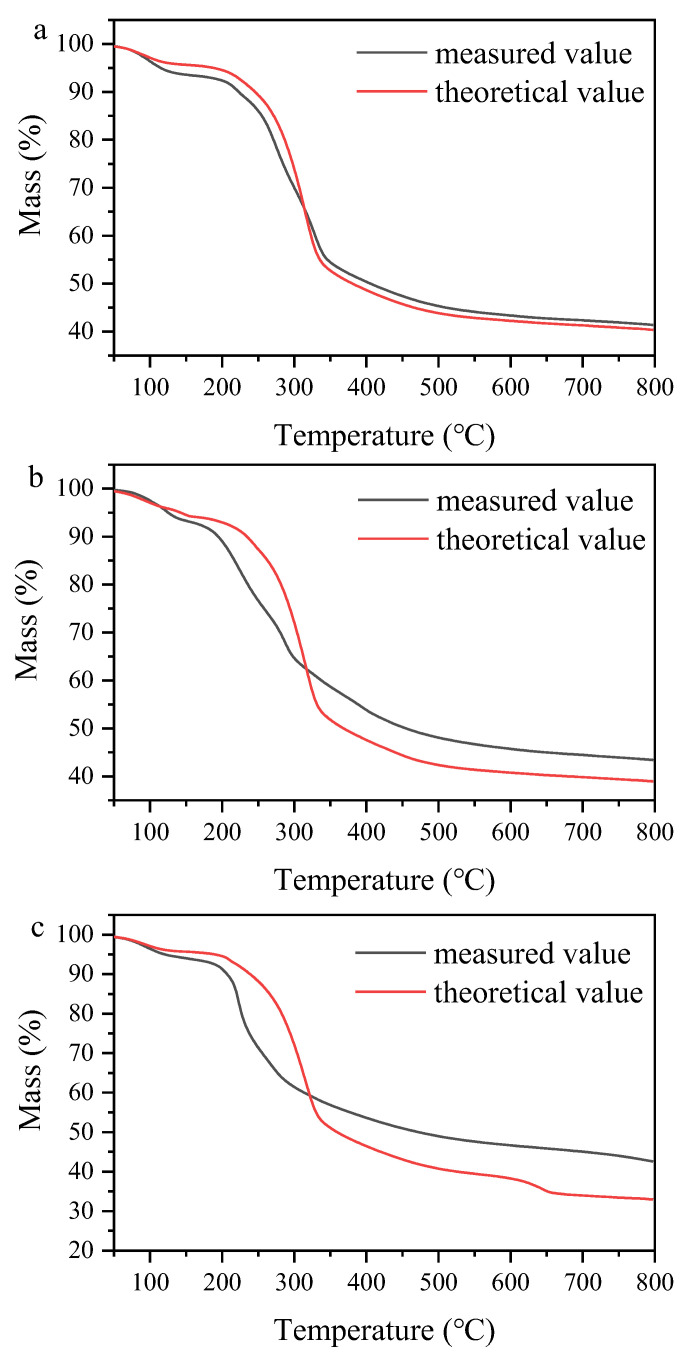
The comparison between experimental and theoretical values of TG ((**a**) CS+KH_2_PO_4_, (**b**) CS+Ca(H_2_PO_4_)_2_, and (**c**) CS+NH_4_H_2_PO_4_).

**Figure 3 molecules-28-03950-f003:**
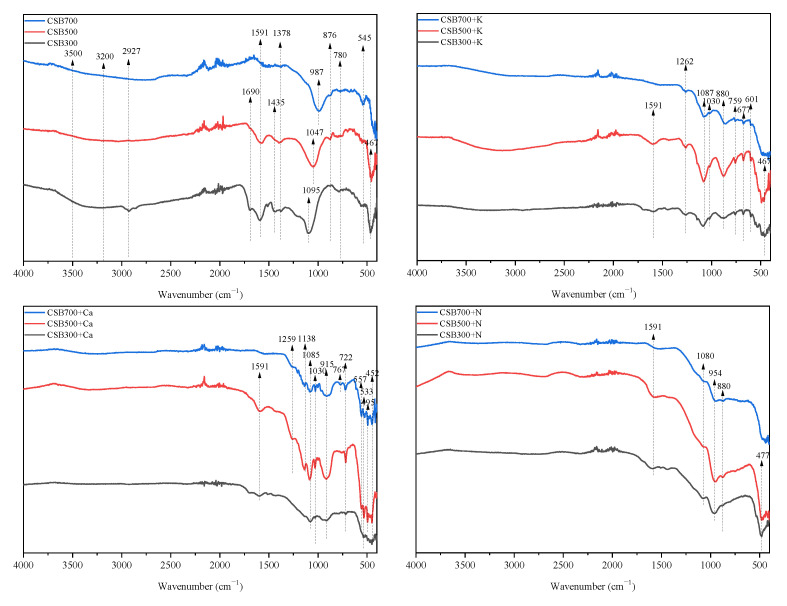
FTIR spectra of biochars.

**Figure 4 molecules-28-03950-f004:**
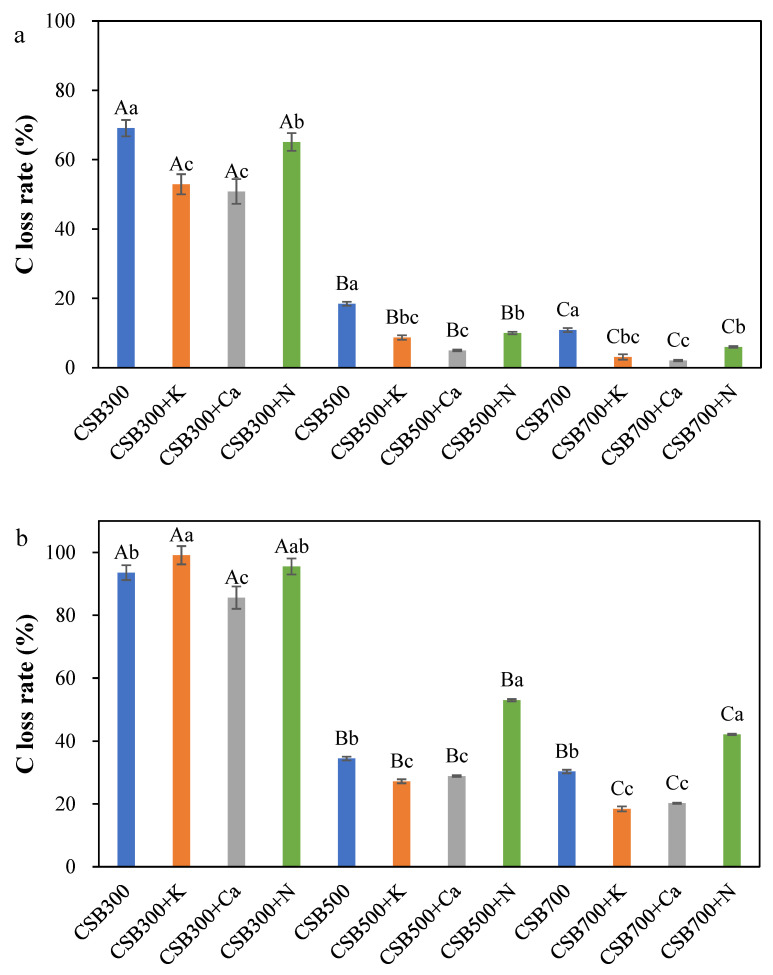
The chemical stability of C in biochar ((**a**) C oxidized by H_2_O_2_, (**b**) C oxidized by K_2_Cr_2_O_7_, (**c**) the total C species). The differences among different temperature treatments are represented by a big letter, and the differences among different phosphates additives are represented with a small letter.

**Figure 5 molecules-28-03950-f005:**
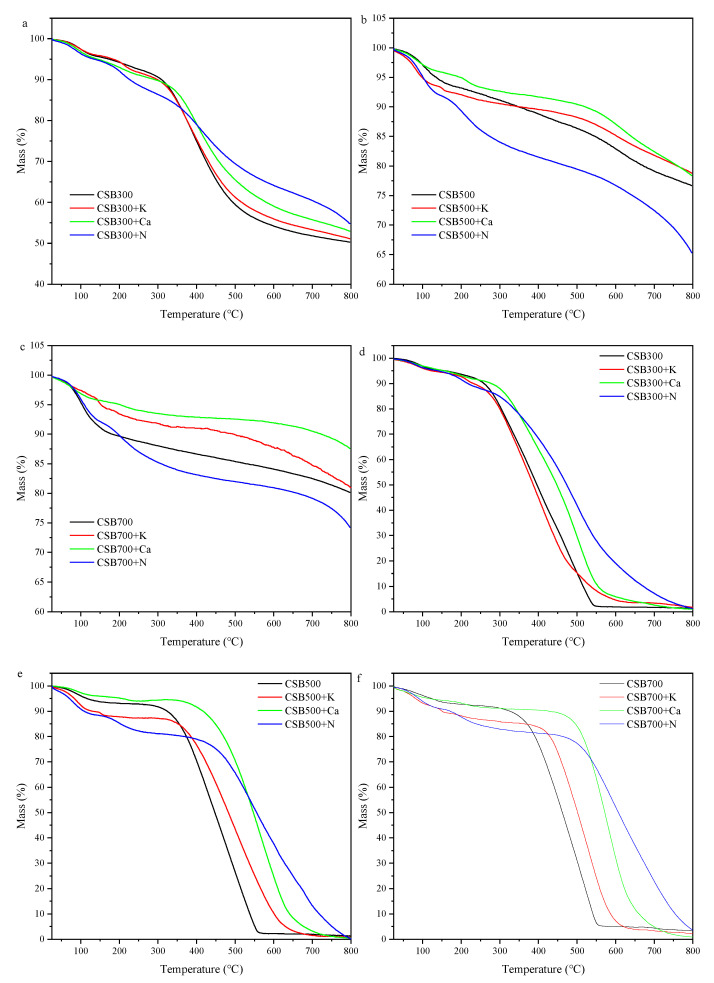
TGA curves of biochars under N_2_ ((**a**) 300 °C, (**b**) 500 °C, and (**c**) 700 °C) and air atmosphere ((**d**) 300 °C, (**e**) 500 °C, and (**f**) 700 °C).

**Figure 6 molecules-28-03950-f006:**
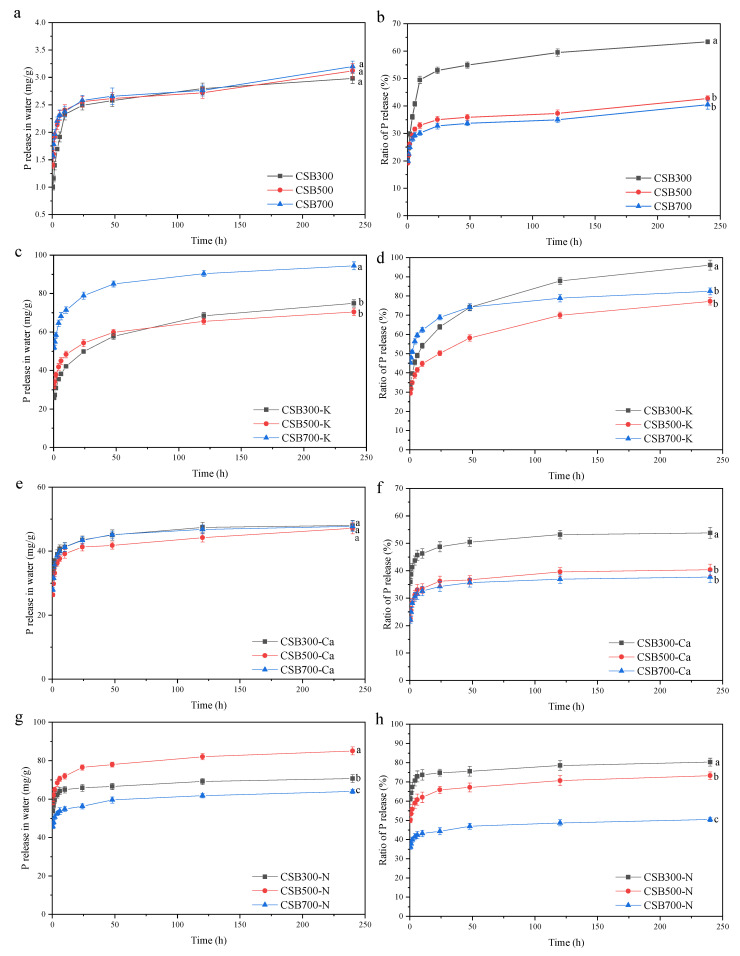
Cumulative amount of released P in water from biochar and its relative ratio to total P over a 10-day (**a**,**c**,**e**,**g**) amount of released P; (**b**,**d**,**f**,**h**) ratio of released P to total P). The differences among different temperature treatments on the 10th day are represented by a small letter.

**Table 1 molecules-28-03950-t001:** The physical and chemical properties of biochar.

	pH	Yield Rate (%)	Ash Content (%)	FC (%)	VS (%)	C (%)	N (%)	O (%)	H (%)	P (%)	H/C
CS	6.95 ±0.06	-	11.35 ±0.20	6.96 ±0.06	81.69 ±0.79	41.45 ±0.26	1.25 ±0.03	15.26 ±0.09	5.67 ±0.06	0.24 ±0.00	1.64 ±0.02
CSB300	7.51 ±0.08 Ca	54.41 ±0.87 Ab	16.98 ±0.11 Bd	40.75 ±0.19 Ba	42.27 ±0.28 Aa	56.10 ±0.31 Ba	1.82 ±0.02 Ab	20.40 ±0.13 Ac	4.35 ±0.05 Aa	0.47 ±0.01 Bc	0.93 ±0.01 Abc
CSB300+K	6.64 ±0.05 Bb	62.24 ±0.77 Aa	39.20 ±0.12 Ba	28.06 ±0.12 Bc	32.74 ±0.36 Ab	40.86 ±0.25 Ab	1.51 ±0.03 Ac	24.01 ±0.20 Ab	3.21 ±0.07 Ab	7.33 ±0.07 Bb	0.95 ±0.02 Ab
CSB300+Ca	3.55 ±0.07 Bc	62.68 ±0.28 Aa	35.88 ±0.25 Bb	32.94 ±0.20 Bb	31.18 ±0.40 Ab	41.38 ±0.21 Ab	1.44 ±0.02 Ac	26.12 ±0.23 Aab	3.05 ±0.02 Ab	8.46 ±0.06 Ba	0.88 ±0.01 Ac
CSB300+N	2.38 ±0.02 Ad	64.67 ±0.47 Aa	27.87 ±0.14 Bc	38.49 ±0.08 Ba	33.64 ±0.21 Ab	42.82 ±0.17 Ab	4.4 ±0.01 Aa	28.17 ±0.19 Aa	4.11 ±0.03 Aa	8.33 ±0.06 Ba	1.15 ±0.03 Aa
CSB500	9.23 ±0.05 Ba	35.13 ±0.19 Bb	26.13 ±0.13 Ad	55.64 ±0.27 Aa	18.23 ±0.19 Bb	60.05 ±0.30 Aa	1.54 ±0.03 Bb	15.95 ±0.12 Bc	2.72 ±0.04 Ba	0.73 ±0.01 Ac	0.54 ±0.00 Ba
CSB500+K	7.25 ±0.06 ABb	45.57 ±0.25 Ba	51.09 ±0.29 Aa	35.4 ±0.16 Ac	13.51 ±0.11 Bc	37.73 ±0.23 Ac	1.02 ±0.02 Bc	21.51 ±0.14 Bb	1.50 ±0.01 Bc	10.17 ±0.07 Ab	0.47 ±0.01 Bb
CSB500+Ca	3.90 ±0.03 Ac	48.40 ±0.20 Ba	46.49 ±0.24 Ab	40.92 ±0.21 Ab	12.59 ±0.09 Bc	42.61 ±0.18 Aab	1.46 ±0.03 Ab	22.08 ±0.20 Bab	1.32 ±0.01 Bd	10.90 ±0.08 Aa	0.37 ±0.01 Bc
CSB500+N	2.42 ±0.04 Ad	49.52 ±0.26 Ba	35.62 ±0.18 Ac	41.02 ±0.22 ABb	23.36 ±0.22 Ba	45.40 ±0.37 Ab	3.57 ±0.06 Ba	23.99 ±0.17 Ba	1.81 ±0.02 Bb	11.18 ±0.10 Aa	0.48 ±0.01 Bb
CSB700	10.17 ±0.11 Aa	32.18 ±0.31 Bb	27.54 ±0.18 Ad	57.06 ±0.16 Aa	15.40 ±0.16 Bab	61.54 ±0.45 Aa	1.27 ±0.04 Cb	10.98 ±0.08 Cc	1.83 ±0.01 Ca	0.79 ±0.03 Ac	0.36 ±0.02 Ca
CSB700+K	7.82 ±0.07 Ab	42.74 ±0.33 Ba	52.73 ±0.30 Aa	35.01 ±0.13 Ac	12.2 6±0.08 Bb	38.85 ±0.37 Ad	0.80 ±0.02 Cc	19.85 ±0.11 Bb	0.96 ±0.01 Cc	10.67 ±0.09 Ab	0.30 ±0.00 Cb
CSB700+Ca	4.04 ±0.03 Ac	44.63 ±0.25 Ba	47.73 ±0.26 Ab	44.92 ±0.14 Ab	7.35 ±0.06 Cc	44.89 ±0.30 Ac	1.24 ±0.02 Bb	21.15 ±0.13 Bab	0.84 ±0.02 Cc	11.88 ±0.11 Aa	0.22 ±0.01 Cc
CSB700+N	2.47 ±0.04 Ad	45.30 ±0.12 Ba	36.99 ±0.09 Ac	45.33 ±0.20 Ab	17.68 ±0.10 Ca	48.00 ±0.24 Ab	3.05 ±0.03 Ca	23.92 ±0.22 Ba	1.33 ±0.03 Cb	11.93 ±0.12 Aa	0.33 ±0.01 Cab

VS: volatile solid; FC: fixed carbon. The differences among different temperature treatments are represented by a small letter, and the differences among different phosphates additives are represented with a big letter.

**Table 2 molecules-28-03950-t002:** Percentage of C1s calculated based on XPS.

	C–C/C=C (284.8 eV)	C–O (C-O-P) (285.9 eV)	C=O (286.8 eV)	O–C=O (288.9 eV)
CSB300	76.92	13.85	6.15	3.08
CSB300+K	75.76	15.15	6.06	3.03
CSB300+Ca	71.94	18.71	6.47	2.88
CSB300+N	64.10	21.79	11.54	2.56
CSB500	82.64	10.74	4.96	1.65
CSB500+K	81.97	11.48	4.10	2.46
CSB500+Ca	81.30	13.01	3.25	2.44
CSB500+N	77.52	15.50	5.43	1.55
CSB700	86.21	7.76	4.31	1.72
CSB700+K	86.96	8.70	3.48	0.87
CSB700+Ca	84.03	10.92	3.36	1.68
CSB700+N	81.97	11.48	4.92	1.64

**Table 3 molecules-28-03950-t003:** Coefficient and constants of models for P-release kinetics from biochar.

	Kinetics Models														
	Pseudo-First			Pseudo-Second			Power Function			Elovich			Parabolic Diffusion		
	R^2^	a	b	R^2^	a	k	R^2^	a	b	R^2^	a	b	R^2^	a	b
CSB300	0.85	2.61	0.34	0.85	2.77	0.19	0.88	1.34	0.16	0.96	3.45	0.71	0.73	1.43	0.12
CSB300+K	0.47	59.18	0.27	0.70	63.14	0.01	1.00	27.78	0.18	0.97	3597.41	2.80 × 10^−4^	0.94	28.37	3.40
CSB300+Ca	0.43	42.66	2.27	-	-	-	0.96	35.22	0.06	0.98	181.83	0.07	0.75	36.19	0.95
CSB300+N	0.44	64.98	3.13	-	-	-	0.94	57.51	−2.57	0.95	1082.39	0.07	0.70	58.84	0.92
CSB500	0.58	2.75	0.78	0.76	2.97	0.34	0.98	1.70	0.14	0.95	4.82	0.78	0.74	1.82	0.09
CSB500+K	0.39	54.99	0.91	0.69	59.74	0.02	0.99	35.16	0.13	0.99	696.78	15.1 × 10^−4^	0.89	36.32	2.56
CSB500+Ca	0.57	40.40	1.58	-	-	-	0.93	31.07	0.08	0.96	87.46	0.04	0.73	32.12	1.13
CSB500+N	0.36	74.42	2.55	-	-	-	0.99	62.03	−4.28	0.99	170.38	0.01	0.82	63.86	1.62
CSB700	0.63	2.50	1.41	0.86	2.63	0.80	0.92	1.88	0.09	0.96	5.28	0.79	0.81	1.90	0.09
CSB700+K	0.34	77.37	1.46	-	-	-	0.99	56.11	0.10	0.99	1415.30	7.36 × 10^−4^	0.85	57.96	2.82
CSB700+Ca	0.62	42.59	1.62	-	-	-	0.90	32.24	0.07	0.94	96.38	0.04	0.67	34.46	1.09
CSB700+N	0.22	56.28	1.73	-	-	-	0.99	48.17	−2.86	0.99	273.24	0.06	0.84	49.30	1.10

## Data Availability

No new data were created.

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
