# Peer review of "Effect of Different Phosphates on Pyrolysis Temperature-Dependent Carbon Sequestration and Phosphorus Release Performance in Biochar"

_molecules, 2023, doi:10.3390/molecules28093950_

Round 1

Reviewer 1 Report

Dear author(s),

there are some inspiring insights throughout the manuscript, and it covers an important topic of potential biomass waste reuse through pyrolysis of impregnated biomass waste with different sources of phosphorus. The final product, biochar could serve as carbon sequestration material as well as soil amendment. The authors revealed the main driving mechanisms related to carbon sequestration and phosphorus release. However, there are few points that needs to be addressed to improve its overall communication. I recommend publication of the manuscript after major correction.

Abstract:

l. 20 – 24: This sentence is too long and difficult to understand, please rephrase.

Introduction:

p. 2, l. 74 – 75: …were selected as additives, which are highly soluble in P … - is this statement correct?

Results and Discussion:

p. 5, Fig 2: Please explain what it is theoretical value.

p.10, l. 259 and other places: the term “pristine biochar” should be properly defined in the Materials and Methods section.

Materials and Methods:

A detailed description of the analytical methods used should be provide, so as any reader could carry out similar measurements, e.g. for XRD measurements what range of scanning angles were used and  other X-ray parameters. Such information should be provided for all analytical techniques used.

p. 15, l. 365: the expression “described minutely” is inappropriate and should be rephrased.

The authors should discuss is a separate section on potential application of the knowledge gained from this project. There are few questions that supposed to be addressed: 1) what are the weaknesses of the methods used, 3) what are the limitations from a commercial point of view? are the lessons learned transferable to other fields? Is impregnation of woody biochar in P-solution economically viable at industrial scale?

Author Response

Abstract:

  1. 20 – 24: This sentence is too long and difficult to understand, please rephrase.

We have rephrased this sentence (Lines 21-25).

Introduction:

  1. 2, l. 74 – 75: …were selected as additives, which are highly soluble in P … - is this statement correct?

We have revised this sentence Line79.

Results and Discussion:

  1. 5, Fig 2: Please explain what it is theoretical value.

The calculation of theoretical value was supplied in Materials and Methods (Lines 352-356).

p.10, l. 259 and other places: the term “pristine biochar” should be properly defined in the Materials and Methods section.

We have added the definition of pristine biochar in Material and Methods (Lines 344).

Materials and Methods:

A detailed description of the analytical methods used should be provide, so as any reader could carry out similar measurements, e.g. for XRD measurements what range of scanning angles were used and other X-ray parameters. Such information should be provided for all analytical techniques used.

We have added more information for all analytical techniques (Lines 376-384).

  1. 15, l. 365: the expression “described minutely” is inappropriate and should be rephrased.

We have rephrased this expression Line 391.

The authors should discuss is a separate section on potential application of the knowledge gained from this project. There are few questions that supposed to be addressed: 1) what are the weaknesses of the methods used, 3) what are the limitations from a commercial point of view? are the lessons learned transferable to other fields? Is impregnation of woody biochar in P-solution economically viable at industrial scale?

We have added a separate section on conclusion (Lines 440-449).

Reviewer 2 Report

This is an interesting work and a comprehensive approach regarding the effect of different phosphates on pyrolysis temperature-dependent carbon sequestration and phosphorus release performance in biochar.

Here are some minor comments:

1.       Page 3: Results in Table 1 are too many to fit, they have changed lines…

2.       Page 4, line 112: Replace ‘were’ with ‘was’.

3.       Page 5, line 144: Replace ‘was’ with ‘were’.

4.       Page 5, line 146: Replace ‘collapse’ with ‘collapsed’.

5.       Page 6, line 167: Rephrase: ‘… was observed in all P-doped biochars …’.

6.       Page 6, line 164: Check paragraph’s order, name… 2.2.4 is the correct title?

7.       Page 6, lines 168-169: ‘… the existence of C-O-P groups in P-doped biochars ..’.

8.       Page 7, line 184: Paragraph 2.2.5 ???

9.   Page 7, line 185: Delete full stop after the word ‘Figure’ in the whole manuscript and replace ‘showed’ with ‘show’.

10.   Page 10, line 252: Paragraph 2.4 ??

11.   Page 11, line 277: Please check… Paragraph 2.5 ???

12.   Page 12, line 305 and Page 13: Please check, it’s Table 3 and not Table 2.

13.   Page 14, line 320: Give reference for methodology.

14.   Page 14, line 325: Replace ‘was’ with ‘were’.

15.   Page 14, paragraphs 3.2 and 3.3: Give references.

16.   Page 14, line 344: Replace ‘were’ with ‘was’.

17.   Check Paragraphs’ titles in the whole text, bold or not? Please follow the same style for fond size…

Author Response

This is an interesting work and a comprehensive approach regarding the effect of different phosphates on pyrolysis temperature-dependent carbon sequestration and phosphorus release performance in biochar.

Here are some minor comments:

  1. Page 3: Results in Table 1 are too many to fit, they have changed lines…

We have readjusted the Table 1.

  1. Page 4, line 112: Replace ‘were’ with ‘was’.

We have revised this error (Line 117).

  1. Page 5, line 144: Replace ‘was’ with ‘were’.

We have revised this error (Line 150).

  1. Page 5, line 146: Replace ‘collapse’ with ‘collapsed’.

We have revised this error (Line 152).

  1. Page 6, line 167: Rephrase: ‘… was observed in all P-doped biochars …’.

We have revised this error (Line 179).

  1. Page 6, line 164: Check paragraph’s order, name… 2.2.4 is the correct title?

We have revised this error (Line 176).

  1. Page 6, lines 168-169: ‘… the existence of C-O-P groups in P-doped biochars ..’.

We have revised this error (Line 180).

  1. Page 7, line 184: Paragraph 2.2.5 ???

We have revised this error (Line 196).

  1. Page 7, line 185: Delete full stop after the word ‘Figure’ in the whole manuscript and replace ‘showed’ with ‘show’.

We have revised these problems in the whole manuscript.

  1. Page 10, line 252: Paragraph 2.4 ??

We have revised this error (Line 264).

  1. Page 11, line 277: Please check… Paragraph 2.5 ???

We have revised this error (Line 289).

  1. Page 12, line 305 and Page 13: Please check, it’s Table 3 and not Table 2.

We have revised this error (Line 329).

  1. Page 14, line 320: Give reference for methodology.

We have added references for methodology (Line 334).

  1. Page 14, line 325: Replace ‘was’ with ‘were’.

We have revised this error (Line 338).

  1. Page 14, paragraphs 3.2 and 3.3: Give references.

We have added references for paragraphs 3.2 and 3.3 (Line 338 and 349).

  1. Page 14, line 344: Replace ‘were’ with ‘was’.

We have revised this error (Line 367).

  1. Check Paragraphs’ titles in the whole text, bold or not? Please follow the same style for fond size…

We have bolded all titles.

Reviewer 3 Report

The topic of the work is interesting. You have shown a large number of results, but quite confusingly. I could not find Supplementary Materials.

Why did you list the results there? the paper is not that big, also some results from the Supplementary Materials should be in the main paper.

Modify the abstract, it looks more like a conclusion.

Section 26, cite the reference, would probably be even more stable if you treated at 800 degrees.

Section 62, what is biochar-based slow?

Why did you choose three kinds of phosphates (KH2PO4, Ca(H2PO4)2, and NH4H2PO4)?

In addition to classic pyrolysis, you could try hydrothermal carbonization

How did you calculate the yield that you stated in the paper and table 1.?

You did not specify the experimental part, what quantity of corn stalks, how it was prepared, you only specified the ratio.

You stated the pyrolysis temperatures, but not the amount of gas, which atmosphere, heating rate, etc.

At TGa, paragraph 2.2.2, you could have done the analysis on 3 speeds and from there do the kinetics

Paragraph 2.2.3 separate SEM and XRD

Once you've done SEM, do EDS

Figure 3. Expand to see the functional groups

Why didn't you do a BET to see the porosity and pore distribution?

You did not specify for each technique the method of how you treated the samples AND what the device models are

Figures. 7 Cumulative amount of released P in water from biochar and its relative ratio to 313 total P over a 10-day, why didn't you give three times, let's say, and some comparison

You could also do ICP from the techniques

You mentioned Raman in the above part of the paper, but I see that you did not show him in the results.

The results are overwhelming and confusing.

You have a lot to correct in order for my opinion to be in favor of accepting your paper in the journal

Author Response

The topic of the work is interesting. You have shown a large number of results, but quite confusingly. I could not find Supplementary Materials.

We have supplied the Supplementary Materials in submission system.

Why did you list the results there? the paper is not that big, also some results from the Supplementary Materials should be in the main paper.

7 Figure and 3 Table were listed in manuscript, which were more important than additional 8 Figure and 1 Table that were presented in Supplementary Materials.

Modify the abstract, it looks more like a conclusion.

We have revised the abstract.

Section 26, cite the reference, would probably be even more stable if you treated at 800 degrees.

We have revised this sentence (Lines 30-31).

Section 62, what is biochar-based slow?

Biochar-based slow fertilizers are fabricated by combining biochar with inorganic fertilizers through various physical or chemical techniques, such as impregnation, pelletizing, and encapsulation (Dong et al., 2021; Wang et al., 2022).

Wang C, Luo D, Zhang X, et al. Biochar-based slow-release of fertilizers for sustainable agriculture: A mini review. Environmental Science and Ecotechnology, 2022: 100167.

Dong D, Li J, Ying S, et al. Mitigation of methane emission in a rice paddy field amended with biochar-based slow-release fertilizer. Science of The Total Environment, 2021, 792(8):148460.

Why did you choose three kinds of phosphates (KH2PO4, Ca(H2PO4)2, and NH4H2PO4)?

Three kinds of phosphates (KH2PO4, Ca(H2PO4)2, and NH4H2PO4), were selected as additives, due to their high solubility.

In addition to classic pyrolysis, you could try hydrothermal carbonization.

In future, we will try to hydrothermal carbonization.

How did you calculate the yield that you stated in the paper and table 1.?

We have added the calculation of yield in Material and Methods (Lines 358-361).

You did not specify the experimental part, what quantity of corn stalks, how it was prepared, you only specified the ratio.

We have supplied detailed information in Materials and Methods (Lines 338-341).

You stated the pyrolysis temperatures, but not the amount of gas, which atmosphere, heating rate, etc.

We have supplied this information in Materials and Methods (Line 343).

At TGa, paragraph 2.2.2, you could have done the analysis on 3 speeds and from there do the kinetics.

The heating rate (10℃/min) of TG was selected as being consisted with the production of biochar.

Paragraph 2.2.3 separate SEM and XRD

We have separated the results of SEM and XRD.

Once you've done SEM, do EDS

We did not make the analysis of EDS. However, major elements including CNHO and P were analyzed by other methods.

Figure 3. Expand to see the functional groups

It is very difficult to expand 12 FTIR spectra in one Figure.

Why didn't you do a BET to see the porosity and pore distribution?

We have analyzed the porous structural of biochar pyrolyzed at 500℃ (Table S1).

You did not specify for each technique the method of how you treated the samples AND what the device models are.

We have added reference for the method of treating samples and device models (Lines 333, 340, 342 and 343).

Figures. 7 Cumulative amount of released P in water from biochar and its relative ratio to 313 total P over a 10-day, why didn't you give three times, let's say, and some comparison.

We have added standard deviation in Figure 7, and the ANOVA analysis of different temperature treatments at 10th day have been also supplied.

You could also do ICP from the techniques

The content of P was measured by ICP (Line 412).

You mentioned Raman in the above part of the paper, but I see that you did not show him in the results. The results are overwhelming and confusing.

The original Raman spectrum and the calculated ratio of ID/IG was showed in supporting information (Figure S5).

You have a lot to correct in order for my opinion to be in favor of accepting your paper in the journal.

Round 2

Reviewer 1 Report

Accept 

Reviewer 3 Report

Dear authors,

you adequately responded to my remarks and suggestions. Because of everything shown, I give a positive opinion that your work will be accepted in the journal